# A Review of the Extruder System Design for Large-Scale Extrusion-Based 3D Concrete Printing

**DOI:** 10.3390/ma16072661

**Published:** 2023-03-27

**Authors:** Hao Chen, Daobo Zhang, Peng Chen, Ning Li, Arnaud Perrot

**Affiliations:** 1Department of Mechanical Engineering, School of Engineering, University of Michigan, Ann Arbor, MI 48109, USA; 2Department of Civil Engineering, School of Engineering, Tsinghua University, Beijing 100190, China; 3Department of Architecture and Built Environment, School of Engineering, University of Nottingham, Nottingham NG7 2RD, UK; 4Department of Mechanical, Aerospace and Civil Engineering, University of Manchester, Manchester M13 9PL, UK; 5Institut de Recherche Dupuy de Lôme, Université Bretagne Sud, 56100 Lorient, France

**Keywords:** concrete extrusion, 3D concrete printing, ram extrusion, extruder system design

## Abstract

Extrusion-based 3D concrete printing (E3DCP) has been appreciated by academia and industry as the most plausible candidate for prospective concrete constructions. Considerable research efforts are dedicated to the material design to improve the extrudability of fresh concrete. However, at the time of writing this paper, there is still a lack of a review paper that highlights the significance of the mechanical design of the E3DCP system. This paper provides a comprehensive review of the mechanical design of the E3DCP extruder system in terms of the extruder system, positioning system and advanced fittings, and their effects on the extrudability are also discussed by relating to the extrusion driving forces and extrusion resistive forces which may include chamber wall shear force, shaping force, nozzle wall shear force, dead zone shear force and layer pressing force. Moreover, a classification framework of the E3DCP system as an extension of the DFC classification framework was proposed. The authors reckoned that such a classification framework could assist a more systematic E3DCP system design.

## 1. Introduction

The traditional formwork-casting method inherited from the ancient Romans underpins the foundation of modern concrete construction. However, the shortcomings of the method have been acknowledged with centuries of practice. Because of its inability to fulfill the increasing structural, sustainable, economic, social and aesthetic requirements, the concrete industry has begun to explore candidate technologies that could revolutionize concrete construction. Buswell et al. [1] outlined a classification framework for the feasible digital fabrication of concrete (DFC) technologies, as shown in Figure 1. 3D concrete material extrusion—referred to as extrusion-based 3D concrete printing (E3DCP) in this paper— is a subclass of 3DCP and has been appreciated by academia and industry as the most plausible candidate for prospective concrete construction. Its commercialization potential has been well-validated in various industrial projects undertaken by construction companies such as XTree [2], COBOD [3], WASP [4], and Sika [5]. Notice that sometimes equivalence is drawn between E3DCP and 3D concrete printing (3DCP), which should be avoided, as the latter is more appropriately referred to as “additive” according to the classification of [1]. In addition, the scope of E3DCP inherently excludes injection 3D concrete printing [6], smart dynamic casting [7], and shotcrete 3D concrete printing [8].

According to [1], any DFC technology can involve a complex process chain within which a single principal process (i.e., shaping or assembly) and a series of sub-processes (i.e., an indispensable process that occurs while executing the principal process) can be identified. In the case of E3DCP, the principal process is the shaping, which consists of the extrusion and deposition processes. However, the sub-processes of E3DCP are more difficult to generalize, as various customizable fittings can be adapted to the E3DCP mechanical system. Based on the sub-processes outlined by [9] and extensive reviews of the literature, the authors have recognized two categories of sub-process for E3DCP: (1) basic sub-processes: those inherited from the traditional formwork-casting process, including the mix proportioning, primary mixing, transport/pumping and curing processes; and (2) advanced sub-processes: those requiring advanced fittings to improve the printing quality or augment the functionality of E3DCP, including the secondary mixing, setting-/fluid-on-demand, in-process reinforcement, interlayer bonding enhancement, finishing, support placement and monitoring and feedback processes.

While the concrete research relating to the basic sub-processes is abundant, the research relating to the principal process and the advanced sub-processes is scarce due to the fact that they are rarely applied to traditional concrete construction projects [10]. With the advent of E3DCP, more research interest has been paid to these two topics in this recent decade. Considerable research efforts are dedicated to the material design (e.g., water-to-cement ratio) of E3DCP, and there have been several prominent review papers [11,12,13,14] that summarize the insights in this regard. However, at the time of writing this paper, there is still a lack of a review paper that highlights the significance of the mechanical design (e.g., nozzle shape, nozzle diameter) of E3DCP.

The complex process chain of E3DCP inevitably entails sophisticated mechanical systems, as shown in Table 1. The purpose of this paper is to provide a comprehensive review of the mechanical systems of the principal process and advanced sub-processes for E3DCP applications. The mechanical systems of basic sub-processes are not included since they are well-established in the concrete industry through decades of practice. Therefore, this paper only concerns the printing system (for the principal process) and advanced fittings (for the advanced sub-processes). The printing system consists of two components: (1) the extruder system, also known as the printhead or manipulator, which performs the extrusion action.; and (2) the positioning system, which enables the deposition action (i.e., extruder movement). Advanced fittings can be added to the printing system to introduce additional advanced sub-processes into the E3DCP process chain.

The rest of the paper is arranged as follows: Section 2 presents the extrusion process and the mechanical design of the extruder system; Section 3 presents the deposition process and the mechanical design of the positioning system; Section 4 presents the advanced sub-processes and the respective advanced fittings; Section 5 analyzed the process chain of E3DCP systems and outlined an E3DCP classification framework as an extension to the classification framework of [1] based on Section 2 and Section 3; and Section 6 presents the conclusion.

## 2. Extrusion Process and Extruder System

The extrusion process is a crucial process of E3DCP wherein the concrete undergoes plastic deformation by passing through an outlet to obtain the desired cross-section profile [15]. To ensure a successful extrusion process, the process requirement of extrusion (i.e., extrudability, which describes the capability of fresh cementitious paste (FCP) to be extruded smoothly throughout the outlet without considerable cross-sectional deformation and with an acceptable degree of splitting/tearing of filament [12,16]) has to be fulfilled.

Depending on the material design, mechanical design (i.e., extruder system design), and operational design, different extrusion behaviors can be observed. A general extruder system is shown in Figure 2, which consists of (1) a piston (ram extrusion mechanism); (2) an axis-symmetric chamber with a diameter of D_c_ and a length of L_c_; and (3) a nozzle (outlet) with a diameter of D_n_ and a length of L_n_. The studies [17,18,19,20] allow the authors to generalize the extrusion behavior of FCP using such a ram extruder system. To enable a successful extrusion, the extrusion driving force, F_e_ (in this case, ram extrusion force, Fram), has to overcome extrusion resistive forces that are responsible for the extrusion pressure drop [21,22,23], which may include: (1) the chamber wall shear force F_cf_ (or friction force) in the billet zone; (2) the shaping force F_pl_ in the shaping zone, also known as the die entry pressure, which is responsible for the plastic deformation of FCP between the chamber and outlet; (3) the nozzle wall shear force F_nf_ (or friction force) in the shaping zone, also known as die land pressure; (4) the dead zone shear force F_df_ (or friction force) in the dead zone; and (5) the layer pressing force, F_lp_ needs to be taken into account when the layer pressing extrusion mode is adopted.

The presence and magnitude of these extrusion forces, which are dependent on the material design, mechanical design, and operational design of the extruder system, can affect the extrusion behavior (e.g., shearing, consolidation and phase separation and dead zone formation), thereby determining the extrusion pressure and the fulfillment of the extrudability requirements [24]. The following section presents the effects of general extruder design, chamber design and outlet design on extrudability.

### 2.1. General Extruder System Design

#### 2.1.1. Extrusion Mechanism

The extrusion mechanism provides extrusion driving forces for concrete extrusion. It describes how FCP is extruded from the extruder entry towards the outlet exit [10]. There are three types of extruder mechanisms: (1) primary motivation [24]; (2) ram extrusion; and (3) screw extrusion, see Figure 3 and Table 2. With the current state-of-the-art of E3DCP, the most common mechanism is the primary motivation, which relies on the pumping and gravity force, F_pg_ to drive the FCP extrudate. Although this approach obviates the installation of additional hardware, thus reducing the associated cost, one of its drawbacks is that the extrusion process is not decoupled from the pumping process. In other words, the extrusion and pumping processes are controlled via identical operational parameters (i.e., pumping pressure/flow rate/extrusion pressure/extrusion rate) so that one does not have specific control over the extrusion process. In addition, due to the low user controllability, the primary motivation cannot provide effective flow regulation when a discontinuous printing path is required (e.g., in the case of support placement). Consequently, the inertia is likely to induce over-extrusion at stop positions as concrete material is ‘pulled’ from the extruder system by gravity.

A substitutive mechanism is the ram extrusion, which relies on repetitive piston movements (driven by F_ram_) to expel the FCP extrudate [25]. The adoption of this mechanism is fairly limited for E3DCP, partially due to the fact that the discontinuity of the ram extrusion could lead to material heterogeneity and defects at the later-on deposition stage [26]. Nonetheless, this mechanism indeed allows higher controllability using the piston descending speed. For printing without accelerator addition at the printhead (so-called 1K approach [9]), the ram extrusion can be adopted to control the timeline from material preparation to material extrusion. Additionally, the ram extrusion mechanism facilitates extrudability characterization, as currently the most-used characterization test (i.e., the ram extrusion test [27]) has identical working principles, which allow for more accurate data acquisition. However, the ram-extrusion process has high-pressure gradients between the ram and the die, so it is more likely to induce consolidation and heterogeneity, especially for high-yield stress fluids [28]. This approach has been adopted by [29,30], and the former has successfully constructed a large-scale power distribution substation with dimensions (l × w × h) of 12.1 × 4.6 × 4.6 m.

The screw extrusion mechanism relies on the use of a screw to impose a screw extrusion force, F_screw_ to continuously convey the FCP towards the outlet [31]. It not only enhances the controllability of the extrusion process (e.g., screw speed control) but also provides some extent of FCP homogenization within the extruder and inhibits the formation of the dead zone [21,32]. Moreover, in this system, the repartition of the extrusion effort along the screw length reduces the risk of consolidation. The screw mechanism is pervasive in the polymer extrusion industry and has started to gain popularity in FDM 3D printing. Multitudinous pioneering FDM 3D printer prototypes integrated with the screw mechanism have been proposed to aid the polymer extrusion process, including the auger screw for multi-material extrusion [33], the conical screw for better extrusion and mixing efficiency [34] and the air-compaction screw [35]. Additionally, some innovative dynamic mixers have also been proposed in other fields, which can be also introduced into E3DCP extruders. For instance, Ishida [36] introduced a mixing blade that could be heated up through electric coils to aid the mixing of high-viscosity solutions. Such a system is particularly relevant for E3DCP as it is well-admitted that the temperature change could prominently affect the FCP hydration progress, thereby altering its extrudability [37].

Nonetheless, due to the nascency of E3DCP, the compatibility of these extrusion mechanisms with different concrete materials (e.g., fiber-reinforced concrete) has not been profoundly explored. For example, most 3D concrete printers with screw mechanisms [38,39,40,41] did not investigate the effects of different screw design options (screw speed, screw type and flight width) on the extrudability [42]. To the authors’ knowledge, the studies [43,44] are the only literature that investigates the screw design for E3DCP applications wherein the effects of screw rotation speed on the buildability of printed concrete specimens were investigated. It was found that too high a rotation speed could induce excessive friction heat and lead to lower fluidity and impaired buildability. In addition, different extrusion mechanisms should be investigated from the energy consumption standpoint, which is necessary for large-scale commercialization. Additionally, although the numerical modeling of concrete extrusion through primary motivation has recently gained profound development [45,46], it remains difficult to model the competition between F_screw_ and the extrusion resistive forces aforementioned for concrete extrusion through screw extrusion. In this case, assessing the environmental impact of the process itself can be required in order to compare it to the material impact [47].

#### 2.1.2. Extruder Wall Roughness

The extruder material largely determines the wall roughness, which in turn affects the friction value at the chamber wall (i.e., F_cf_) and at the nozzle wall (i.e., F_nf_), see Table 3 [10]. However, rarely do E3DCP articles mention this aspect, thus overlooking the significance of wall roughness. A preliminary work carried out by [48] investigated the impacts of extruder walls with different surface roughnesses, Ra, on the friction stress. As expected, a rougher surface is associated with a higher dynamic friction coefficient value. The dynamic friction coefficients were found to be 0.49, 0.59 and 0.87, respectively, for extruder walls with Ra of 0.52 µm, 2.77 µm and 100 µm. Syahrullail et al. [49] studied the polishing effects on the extrusion process of metal. Two extruders are subjected to polishing at different regions: extruder 1 is subjected to polishing at the whole extruder wall surface to achieve Ra of 0.05 µm, and extruder 2 is only subjected to polishing at the nozzle wall surface to achieve Ra of 0.05 µm. It was found that extruders 1 and 2 are associated with extrusion loads of 48 kN and 57 kN, respectively. One can expect that a lower F_cf_ is associated with a lower extrusion pressure, which reduces the risk of phase separation and is also beneficial from the economic point of view.

To improve the surface roughness and reduce F_cf_, apart from using different materials or polishing, one can apply lubricants onto the surface, which is akin to the application of priming in assisting concrete pumps [50]. For example, Syahrullail et al. [51] applied vegetable oil to improve the surface roughness of metal extruder walls. Therefore, it is promising to investigate the compatible priming materials to realize a smoother E3DCP extrusion process of high-viscosity FCP. Further studies are necessary to investigate the coupling effects of wall roughness (e.g., by using different wall materials or applying polishing) and different concrete materials (e.g., low- and high-viscosity) on the extrusion phenomena (e.g., consolidation) and extrudability.

### 2.2. Chamber Design

#### 2.2.1. Chamber Number

The extruder chamber, sometimes referred to as the extruder barrel or hopper, is where the pumped FCP arrives during the extrusion [10]. In general, most E3DCP extruder systems are equipped with a single chamber, but there are extruder systems with no chamber to reduce the equipment cost [52] or with double chambers [29], see Figure 4a. An extruder system without a chamber, when combined with primary motivation, is particularly unfavorable for flow regulation, as there is no buffering of the “gravity pulling” of concrete materials. Ji et al. [29] adopted the double chamber to resolve the discontinuity issue of the ram extrusion mechanism that could unnecessarily increase the printing duration. In this manner, while one chamber is ram-extruding FCP, the other chamber is closed for the preparation of FCP, which can be ready when the first chamber finishes extrusion, thereby achieving uninterrupted ram-extrusion [29]. Additionally, the double chamber could realize the possibility of multi-material extrusion, thereby realizing the functionally graded material as suggested by [53]. Moreover, a variation of the double chamber called Y-shaped chamber was recently proposed [54]. The approach enables simultaneous pumping of two fluid materials in two separate chambers and intermixing in a final extrusion chamber, leading to a fast-setting material. Nonetheless, except in the case of functionally graded materials, the homogeneity of concrete material is always a crucial concern for E3DCP. This means that adopting more than one chamber for extrusion requires the rigorous controls of the mix consistency and the time-lapse of extrusion from different chambers to ensure minimal variations of material rheological properties.

#### 2.2.2. Chamber Diameter and Length

There have been fairly limited studies concerning the effects of L_c_ and D_c_ on the extrudability of FCP. Vallurupalli et al. [31] reported that D_c_ influences the shear rate imposed on FCP, which in turn affects the extrusion behavior of FCP. A larger D_c_ corresponds to a lower shear rate and a lower F_pl_ according to the Benbow and Bridgewater model [55], which likely induces an extrusion mode close to infinite extrusion mode, while a smaller D_c_ likely leads to a free flow extrusion mode [31]. Nonetheless, the clear origin of the increased shear rate in terms of extrusion forces requires more investigation. Vallurupalli et al. [31] also reported that a greater L_c_ favors the shear-induced particle migration phenomenon and formation of the lubrication layer. Investigation of the correlation between the extrusion mode with D_c_, L_c_ and concrete mix designs (e.g., solid concentrations, fiber reinforcements) and the establishment of a standard guideline can aid the mechanical engineers to decide the initial chamber dimensions based on the desired extrusion mode and pre-specified concrete mix designs. Furthermore, the chamber dimensions may also influence the fiber alignment effect [56], therefore, follow-up research is also necessary in this topic in light of the increasing popularity of fiber-reinforced concrete and engineering cementitious materials for E3DCP applications.

#### 2.2.3. Chamber Tapering

The chamber tapering describes how the chamber is connected to the outlet: (1) without tapering: abrupt contraction with an outlet entry angle, θ_e_ of 90°; and (2) with tapering: progressive contraction with a θ_e_ less than 90°, see Figure 4b and Table 4. The work carried out by O’Neill et al. [57] on the extrusion of calcium phosphate paste (non-Newtonian fluid) can provide a reference for the chamber tapering effects during concrete extrusion. The study investigated the relationship between the outlet entry angle and the extent of phase separation (reflected by the liquid powder ratio (LPR) of the extrudate). Extruders with θ_e_ of 90° (without tapering), 55° and 45° are used for extrusion tests, and the LPRs of the extrudate are 0.492, 0.449 and 0.434, respectively. The results demonstrated that an acute θ_e_ (45°) can effectively reduce the required LPR value and mitigate phase separation. The authors ascribed the mitigation partially to the low extrusion pressure due to the lower F_pl_ associated with the tapered chamber [17]. Similar results were obtained by [58]. However, Nienhaus et al. [59] found that although a small θ_e_ of 28° indeed induces relatively low extrusion force (around 20.5 N) compared to those of 45° (around 22.7 N) and 70.5° (around 23 N), a too small θ_e_ (i.e., 15.5°) does not necessarily lead to a further decrease in extrusion force (around 21.7 N). This is because the increase in the tapering surface area due to low θ_e_ can lead to a higher F_cf_, thereby raising the extrusion force. This hypothesis still needs more experimental validation. Additionally, another merit of the tapered chamber is that its geometric nature could avoid the formation of dead zones near the outlet, which in turn reduces the blockage risk and improves the extrusion quality [57]. More studies should be conducted to investigate the coupling effects of entry angle and concrete materials of different viscosity and solid concentrations on the extrusion phenomena (e.g., phase separation), from which one can draw the guidelines of entry angles for different concrete materials.

### 2.3. Outlet Design

#### 2.3.1. Outlet Form

The outlet of the extruder system imposes its shape on the FCP during extrusion. It can be in the form of an orifice or a nozzle, as shown in Figure 5a. Although orifice outlet is rarely seen in E3DCP applications, its benefits have been highlighted by [22]. Nair et al. [22] investigated E3DCP extruder systems with orifice and nozzle forms, and it was found that the nozzle increases the required extrusion pressure due to the presence of nozzle wall friction (F_nf_). This result suggests that the extrusion through a nozzle has a slightly higher phase separation than that of the orifice. The study of O’Neill et al. [57] showed that a nozzle outlet could slightly reduce the phase separation; the conclusion contradicts that of [22]. A plausible explanation lies within the difference in the rheological properties of FCP and calcium phosphate paste used in the two studies—the nozzle effect may be more conspicuous when a high-viscosity paste is used.

In some cases, multiple nozzles are employed for the E3DCP application. For example, contour crafting [60] customized a multi-nozzle extruder to simultaneously create the outer contours and the inner skeletons, which enables a more massive deposition process. The Huashang Tengda company also uses a dual nozzle system that can encase a preplaced reinforcement grid [61].

#### 2.3.2. Outlet Orientation

When the outlet is in the form of a nozzle (which is usually the case for E3DCP), outlet orientation can be vertical, horizontal [62], or tilted [30,63], see Figure 5b. Since investigations regarding the effect of outlet orientation on extrudability are fairly scarce, no conclusive statements could be made. The vertical orientation is the default option for most E3DCP extruder systems. It is also a reasonable option to implement the layer pressing strategy where the printhead perpendicular to the concrete layers can directly apply the nozzle pressure to control the layer thickness [64]. By inspection of the study by [62], it is logical to hypothesize that the horizontal outlet orientation particularly suits the infinite brick extrusion mode because the horizontal forming process that ensues the gravitational fall of FCP could improve the geometry accuracy. Ramakrishnan et al. [63] experimented with tilted nozzles with angles of 45°, 60° and 90° between the nozzle and the print bed for hollow-filament extrusion, and it was found that tilted angles of 45° provided the optimum extrudability and buildability. The titled nozzle is also of interest for printing complex shapes with cantilevers inspired by the old masonry [65], in which case a more complex six-axis robotic system is required [66].

#### 2.3.3. Outlet Tapering

In the case of the nozzle, the outlet tapering has also to be notified as it has considerable effects on the extrusion pressure [22]. It basically describes the change in nozzle inner diameters from entry to exit: (1) without tapering: uniform nozzle where the nozzle entry diameter, D_entry_ and nozzle exit diameter, D_exit_ are the same; and (2) with tapering—non-uniform nozzle—where D_entry_ and D_exit_ are different, as shown in Figure 5c. Nair et al. [22] demonstrated that the tapered nozzle N10-4 (i.e., nozzle diameter of 10 mm tapering to 4 mm) requires a higher extrusion pressure compared to the uniform nozzle N10-10. This is partially due to the higher surface area of the tapered nozzle, which generates a higher shear force, F_nf_. This finding is in line with the claim of [31], that for an extruder with complex nozzle geometry, such as the tapered nozzle, a considerable portion of the extrusion pressure originates from the shearing in the nozzle.

#### 2.3.4. Outlet Cross-Section Shape

From the extrusion mode standpoint, a rectilinear nozzle is more suitable for stiff FCP extrusion, as the extrudate from the circular nozzle will be subjected to a considerable deformation before reaching equilibrium [67,68]. Some studies have affirmed the merits of rectilinear cross-shapes for the mechanical properties of FCP extrudate after extrusion. For instance, Kwon et al. [69] utilized the finite element analysis method to examine the deposition mechanisms of the E3DCP process and revealed that a square cross-section can create an excellent surface profile and favors interlayer bonding. Shakor et al. [70] also confirmed that the square/rectangular cross-sections produce filaments with a higher surface area, which allows the vertical forces to be more distributed, thereby increasing the failure layers.

Liu et al. [71] have investigated the flow behavior during extrusion and deposition at corners using rotational rectangular nozzles with different aspect ratios. A mass distribution ratio, Φ is proposed to characterize the flow of FCP extrudate at corners, which is essentially the ratio of the inner side cross-section area, S_i_ to the outer side cross-section area of the FCP extrudate, S_o_ as shown in Figure 6. It was shown that a rectangular nozzle can lead to non-uniform mass distribution at corners due to different deposition rates at the inner and outer radius. The result further reveals that, by decreasing the aspect ratio of the rectangular nozzle, Φ decreases, corresponding to a more uniform mass distribution. Accordingly, one can recognize that the circular cross-section more likely induces a free flow extrusion mode, whereas the rectilinear cross-section more likely induces an infinite brick extrusion mode.

In consideration of the cost and technical complexity, a circular cross-section is beneficial as it obviates the rotation mechanism, which would be otherwise necessary for a rectilinear cross-section as an extra degree of freedom to control the facing direction of the outlet [67].

Apart from the relatively simple regular circular and recti-linear cross-sections, more sophisticated geometries, although associated with higher capital costs and technical complexity, have been introduced to realize more complex structures. Ramakrishnan et al. [63] proposed hollow-core and U-shape cross-sections to extrude hollow filaments for printing lightweight structures. Lao et al. [72] established a machine learning-based artificial neural network model (MLANN) based on various process parameters (including the outlet cross-section shape) to predict the extrudate geometry, and the high correspondence between the predicted and experimental extrudate geometry allowed the authors to use the trained model to optimize the outlet cross-section shape, thereby improving the printing quality of printed parts. In continuation, the same group [73] developed a flexible nozzle that could vary the cross-section shape to dynamically control the extrudate geometry during E3DCP to eliminate the staircase effect, thereby improving the surface finish of the FCP extrudate. Concretely, according to the designated CAD model, the MLANN model can predict the suitable nozzle cross-section shape for each layer, based on which the flexible nozzle can dynamically respond and produce an extrudate of the desired geometry. This extruder design resonates with the initiative of [62], who devised an extruder system that can extrude FCP with different aspect ratios using a replaceable modular nozzle or an adjustable-geometry nozzle. The former requires manual replacement during extrusion, while the latter allows for a more automated extrusion.

#### 2.3.5. Outlet Size

Duballet et al. [74] used the layer height to characterize the extrusion scale of an E3DCP system. However, the outlet size may be a better option, because different layer heights can be achieved with the same outlet size through layer pressing, as shown by [75]. The outlet size can be represented as outlet entry size, D_entry_ and outlet exit size, D_exit_ (D_entry_ = D_exit_ for orifice and non-tapered nozzle and D_entry_ > D_exit_ for non-tapered nozzle), see Figure 7. For a square/rectangular cross-section, the width (i.e., width) is adopted, whereas, for a circular cross-section, the diameter is considered.

Studies relating to calcium phosphate extrusion are excellent references to infer the effects of outlet size on the properties of FCP extrudate. For example, Burguera et al. [76] found an inverse relationship between the extrusion force of calcium phosphate and the nozzle cross-section area. The finding is reasonable according to the Poiseuille equation [77] and justifies the claim that a smaller nozzle cross-section corresponds to enhanced shaping force, F_pl_ and nozzle wall shear force F_nf_. Nair et al. [22,78] observed a similar proportional relationship between the extrusion pressure and outlet size for the extrusion of cement paste. The study of O’Neill et al. [57] reached a seemingly contradictory conclusion with the previous literature that varying the outlet size has a marginal effect on the extent of phase separation. It is pointed out by the authors that the contradiction arises from the difference in viscosities of the pastes used in the studies. By comparing the results of [22] with [57], one can infer that the effect of outlet size on the extrusion quality is more pronounced in high-viscosity materials (for example, the FCP for E3DCP). Nonetheless, more experimental studies are required to justify the hypothesis.

Outlet size also influences the likelihood of blockage. Concretely, the selection of the outlet size should take account of the solid particle size within the concrete mix. As can be seen in Table 5, most E3DCP studies adopted a B_E_ (i.e., ratio of Dexit to maximum aggregate size [79]) above 5. According to [80], the maximum aggregate size should be smaller than 1/3 of the Dexit to prevent outlet blockage. Cheikh et al. [79] suggested that B_E_ should be greater than 4.25. Similarly, if the concrete mix incorporates fiber as reinforcement, it is important to consider the compatibility in terms of the minimum D_exit_. This criterion could be used in conjunction with the maximum D_exit_ suggested by [56] if fibers are incorporated as the reinforcement. Arunothayan et al. [56] found that a smaller nozzle diameter (in this case < 20 mm for the range of 10, 15, 20, 30, 40 mm) could induce more effective elongation flow and favor the fiber alignment parallel to the extrusion direction during the extrusion process, thereby enhancing the mechanical properties, such as the modulus of rupture and flexural strength in Z and Y directions. The study confirmed the presence of an upper bound on the ratio of nozzle diameter to fiber length above which the fiber alignment effect during extrusion becomes independent of the nozzle diameter [56].

Having acknowledged the pronounced influence of D_exit_ on various aspects of E3DCP, the concept of the variable-size nozzle has been designed to add an extra degree of freedom to fully exploit the geometric freedom of E3DCP. While Mechtcherine et al. [62] suggest a modular approach, Xu et al. [81] have developed an extruder system with a variable-size square nozzle. Such an approach could increase the adaptability of the concrete printers and allow the printing of designs with high geometrical complexity. Two curved prototypes were printed, based upon which the authors recognized that a rectangular cross-section nozzle equipped with a trowel system could further boost the geometrical accuracy of such a variable-size nozzle.

Overall, choosing an outlet size is a manifold task that must consider the extrudability, economic and aesthetic demands. It is well-admitted that, while a large outlet size enables a massive extrusion and shortens the printing duration of the structure, it sacrifices the structure resolution, which is crucial for aesthetic [82]. One also has to keep in mind that outlet size is closely related to the ratio of the printing velocity to the flow rate, which in turn affects the material tearing and over-extrusion [45,83]. The consequences of the outlet size on the extrusion are summarized in Table 6.

**Table 5 materials-16-02661-t005:** Examples of maximum aggregate size, fiber length and D_exit_ adopted in E3DCP literature.

Reference	Maximum Aggregate Size (mm)	Fiber Length (L × D mm)	Outlet Exit Size (D/L × W mm)	B_E_	Standoff Distance (mm)	Extrusion Velocity (mm/s)
[84]	2	-	28 × 18	9	-	-
[85]	4.75	-	25	5.26	-	-
[86]	1.18	-	30 × 15	12.71	-	-
[87]	1.20	-	30 × 15	12.50	-	-
[88]	4	-	25	6.25	-	-
[89]	1.15	-	15 × 7	6.09	-	476.19
[90]	1	-	40 × 10, 25 × 25	10, 25	10, 25	-
[91]	2	-	20	10	-	-
[92]	1.18	12 × 0.0014	13 × 30	11.02	-	-
[93]	2	-	19	9.50	10	-
[37]	1	6	25 × 15	15	15	-
[94]	1.2	-	30 × 15	12.50	15	66.6
[95]	0.9	-	30	33.33	0–10	28.29
[96]	1.15	-	-	-	0, 2, 4	-
[22]	-	-	4, 10	-	-	31.83, 5.10
[25]	-	-	45	-	-	50–120
[97]	0.5	-	20	-	10	47.22
[83]	-	-	20	-	8–20	43.19
[81]	1.2	-	10–24 × 10–24	-	-	-
[98]	-	-	30 × 15	-	-	-
[80]	10	-	30	-	-	35.37
[99]	9.5	-	29.2	3.07	-	422.60
[100]	2	12 × 0.04	30 × 20	10	-	44
[68]	-	-	20	-	-	-
[101]	-	-	20 × 20, 30 × 10	-	-	-
[102]	0.5	6	25	50	-	-
[29]	-	-	-	-	-	-
[38]	1	9 × 0.023	8 × 30	8	-	-
[103]	0.5	-	25	50	7.5–17.5	40
[104]	2	-	-	-	-	-

**Table 6 materials-16-02661-t006:** The effects of outlet size on the extrusion forces, extrusion behaviors, extrudability and economic aspects and technical complexity based on the same material design and operational design.

	Large Outlet Size	Small Outlet Size
Extrusion resistive forces	• Lower F_pl_ and F_nf_	• Higher F_pl_ and F_nf_
Extrusion behaviors	• Lower shearing • Reduce the consolidation and phase separation and dead zone formation • Lower risk of blockage • Lower extrusion pressure required	• Higher shearing • Increase the consolidation and phase separation and dead zone formation • Higher risk of blockage • Higher extrusion pressure required
Extrudability	• More smooth extrusion	• Less smooth extrusion
Economic aspects and technical complexity	• Lower energy consumption • No additional technical complexity	• Higher energy consumption • No additional technical complexity

Studies have also been conducted to examine the effects of L_n_ on extrusion pressure. Burguera et al. [76] and Fatimi et al. [105] found that a greater L_n_ could increase the required extrusion pressure of calcium phosphate paste, thereby increasing F_nf_ and the risk of phase separation. Nienhaus et al. [59] demonstrated that a longer nozzle (or a greater nozzle length-to-nozzle diameter ratio) induces a significantly higher extrusion force during the extrusion of plastic filament. Nair et al. [22,78] obtained a similar conclusion that the extrusion pressure is dependent on the nozzle length-to-diameter ratio. 

## 3. Deposition Process and Positioning System

Deposition is an additive process where a material is layered onto another layer of the material. During the E3DCP deposition process, the most important property is the buildability, which is defined as the ability of the deposited concrete filament to self-support and resist deformation without formworks [12,106]. The buildability stipulates that the concrete filament should provide sufficient resistance against plastic material failure, elastic buckling failure as well as excessive deformation [107]. However, the mechanical design of the positioning system has a relatively insignificant impact on the buildability compared to the material design. Therefore, the following section presents the mechanical design of the positioning system from a more practical perspective.

Four types of E3DCP positioning systems can be identified: gantry system, robotic arm system, delta system, and swarm system. Each category is characterized by a different degree of freedom and build volume.

The gantry system is the most common positioning system for E3DCP applications due to its ease of operational and cost-effectiveness. A gantry system is generally characterized by three DOFs of translational movements in x, y, z directions (Cartesian coordinate), but sometimes an additional rotational DOF can be added at the extruder to have four DOFs [108,109]. The build volume of the gantry system is constrained by the physical dimensions of the supporting frames in x, y and z directions, and it could range from desktop-scale for laboratory purpose to industrial-scale for construction purposes, see Table 7. To overcome the limited dimension of the gantry system, COBOD [3] has developed a flexible-dimension gantry concrete printer, BOD2, which could be assembled from multiple modular units of 2.5 × 2.5 × 2.5 to fit different construction scenarios. The contour crafting company [60] and IconBuild [110] retrofitted the gantry concrete printer with sliding rails to expand the workspace in one horizontal direction. From a practical standpoint, the robustness of the gantry system can sustain the on-site harsh conditions, however, it could be associated with considerable manual works in assembly and disassembly [62]. Additionally, the accuracy and repeatability of gantry printers are sufficient to complete large-scale E3DCP projects but they are not comparable to the robotic arm system.

The robotic arm system printer generally consists of multiple links connecting altogether at rotary joints, which provides the system with more DOFs (six or more) and allows it to print more sophisticated designs. For example, Lim et al. [111] have pointed out that the staircase effect that typically associates with the extrusion-based 3DCP can be mitigated by adopting the curved-layer printing strategy instead of flat-layer printing. Concretely, in this approach, the extruder nozzle is positioned perpendicular to the target surface throughout the extrusion process so that the surface roughness and geometric accuracy can be improved. To fully exploit the potential of this approach, a position system with four or more DOFs is essential. Similarly, Gosselin et al. [112] recommended utilizing a six-axis robotic arm to realize the tangential continuity method for toolpath planning, which could produce non-planar layers with locally varying thicknesses, thereby unleashing the geometrical freedom of E3DCP to a greater extent. The approach has been used to 3D print multifunctional structures such as the thermal insulation wall and acoustic damping wall. Motamedi et al. [113] utilized a six-axis ABB robotic arm to print an overhang structure without support, which is only possible with the capability of the robotic arm to adjust the angle between the nozzle and printing surface.

However, most industrial robotic arms (e.g., Kuka, ABB and Fanuc) have limited workspace. Once set-up, they can only print structures within a pie-shaped zone formed by the arm reach (generally less than 5 m), which will not suffice for conducting large-scale 3DCP projects. There have been various strategies proposed to extend the reach of the robotic arm: (1) installation of the extension arm at the extruder end [114]; (2) lifting of the robotic arms: the construction company Apis Cor [115] employed a crane to lift the pillar-like robotic arm printer after finishing printing tasks at one point; (3) provision of mobility to the robotic arm: the construction company Cybe [116] and the research team from NanYang Technical University [117] have both installed a mobile base underneath the robotic arm to enable theoretically infinite workspace in horizontal direction, and the research team from TU Dresden conceptualized the adaptation of a truck-mounted pump for E3DCP. However, such an approach imposes more strict requirements on the spatial localization of robotic arm, site conditions (e.g., flatness) as well as the weather conditions (e.g., low wind); (4) carrier system: ETH Zurich researchers [104] installed a six-axis ABB IRB 4600 robotic arm on a Güdel 3-axis gantry at ceiling to increase both the horizontal and vertical workspace, and the team from TsingHua University [118] provided an elevator platform to extend the workspace; and (5) multiple robotic arms: Zhang et al. [119] employed two mobile robotic arms to print structure simultaneously. Such an approach requires complex robotic path planning as well as collision checks before printing. Despite the multitudinous benefits of robotic arms, compared to the robustness of the gantry system, the delicacy of the robotic arm system has raised the suspicion of its suitability for rough on-site conditions, which explains why the majority of robotic arm printers are used under off-site conditions [62].

Apart from the mainstream gantry and robotic arm systems, there have been some innovative systems developed for E3DCP applications. For example, the construction company WASP [4] has customized a Delta 3D concrete printer called BigDelta with a dimension of 7 × 7 × 12 m. The printer consists of three cable-arms connected to joints at frame supports, and each arm could move independently in the y-direction, forming a navigation based on the polar coordinate. The German Fraunhofer Institute [120] also developed a similar delta 3D concrete printer based on eight cable arms. The delta system also has dimension constraints within the frame, and it also suffers from a higher risk of collision with the already printed parts compared to the gantry system [62]. The Institute for Advanced Architecture of Catalonia [121] has designed three swarm 3D concrete printers that could work collaboratively to produce structures: (1) the base robot, which deposits the first ten layers of concrete filaments to create a foundation; (2) the grip robot, which rests on the previously bult foundation and continues deposition to finish the structure; and (3) the vacuum robot, which climbs on the surface of the finished structure and deposits concrete filaments in z-direction. Theoretically, without considering the layer cycle time, such a swarm system can be used to construct large-scale concrete structures without dimensional limitation, especially in horizontal directions. Nonetheless, the technology remains relatively nascent and needs more exploration.

**Table 7 materials-16-02661-t007:** Some examples of 3D concrete printers in terms of position system, build volume, horizontal printing speed, layer height and layer width.

Reference	Positioning System	Degree of Freedom	Build Volume (L × W × H m)/Reach (m)
[29]	Gantry	3-axis	20 × 18 × 18
[94]	Gantry	3-axis	1.2 × 1.2 × 1.0
[38]	Gantry	3-axis	0.5 × 0.39 × 1.1
[103]	Robotic arm	6-axis Fanuc R-2000iC/165F	-
[92]	Gantry	3-axis	-
[95]	Gantry	3-axis	3.0 × 3.0 × 3.0
[25]	Robotic arm	6-axis KUKA KR60 HA	-
[80]	Gantry	3-axis	1.8 × 1.8 × 1.5
[90]	Gantry	4-axis	9 × 4.5 × 2.8
[93]	Gantry	3-axis	0.15 × 0.15 × 0.12
[89]	Robotic arm	6-axis Denso	-
[97]	Robotic arm	6-axis FANUC R-2000iC/165F	-
[83]	Gantry	3-axis	-
[104]	Robotic arm and gantry	6-axis ABB IRB 4600 robotic arm hanging on a Güdel 3-axis gantry	-
[99]	Gantry	3-axis	10.36 × 2.74 × 3.05
[100]	Gantry	3-axis	0.40 × 0.30 × 0.30
[101]	Gantry	4-axis	-
[108]	Gantry	4-axis	-
[3]	Gantry	3-axis	Infinite × 14.6 × 8.1
[110]	Gantry	3-axis	Infinite × 8.53 × 2.59
[116]	Robotic arm	6-axis	2.65–3.50
[116]	Robotic arm	7-axis	Infinite × Infinite × ~3
[120]	Delta system	-	17 × 12 × 5
[4]	Delta system	-	7 × 7 × 12 m

## 4. Advanced Sub-Processes and Advanced Fittings

According to the literature the authors have reviewed, the advanced fittings can be classified as the secondary mixing, setting-/fluid-on-demand, in-process reinforcement, interlayer bonding enhancement, finishing, support placement and monitoring, and feedback processes. The inclusions of the advanced sub-processes within the printing system generally increase the energy, machine and maintenance costs (in the passive systems, the energy increase may be insignificant). Additionally, they may increase the energy and material costs as well as the technical complexity of the overall system, as shown in Table 8.

### 4.1. Secondary Mixing Sub-Process and System

In general, the secondary mixing is an optional set-up to remove the pump-induced heterogeneity after primary mixing, and it is particularly essential when the setting-on-demand by secondary dosage [104,122,123] is adopted. The secondary dosage relies on the addition of chemical admixtures and/or sometimes water, cement, and aggregate at the extruder to control the FCP rheology during the extrusion process. As suggested by [124], for a 30 cm long extruder and an average extrusion velocity of 5 cm/s, the admixtures introduced at the extruder experience a residence time of 6 s, which is not sufficient for the effective penetration of additive particles into the FCP according to the Stokes–Einstein equation. The non-uniform distribution of admixtures could adversely affect the rheological as well as mechanical properties of concrete [124]. Therefore, it is essential to implement secondary mixing (e.g., static mixer and dynamic mixer) to shear FCP so that the admixtures particles can be uniformly distributed within a short residence time. This is referred to as the 2K approach by [9] as opposed to the 1K approach (i.e., no secondary mixing).

Static mixers rely on the configuration of stationary blades to alter the flow path of FCP, thereby generating turbulences that could disperse the additive particles. The effectiveness of this approach depends on the blade configuration. While this approach is appraised for its cost-effectiveness, it is also associated with some drawbacks. For instance, the installation of static mixers within a confined extruder could increase the risk of blockage, which is particularly of concern for high viscosity FCP and limits the aggregate size. Ghanem et al. [125] emphasized that static mixers should be chosen according to the considered material. Thakur et al. [126] have summarized the applicability of different static mixers. For example, the Kenics static mixer is suitable for high-viscosity liquid [127]. Nonetheless, static mixers are currently seldom applied for E3DCP scenarios, so their feasibility needs to be further investigated. To the authors’ knowledge, only a few examples of static mixer applications in E3DCP are reported in the literature [52]. Tao et al. [54,128] report that the use of a helicoidal static mixer can lead to unperfect mixing resulting in striation-shaped heterogeneities that require a careful on-purpose design of the static mixer shape and length.

Dynamic mixers rely on the strong shearing generated by the motor-driven blades to achieve dispersion [124]. In comparison to the static mixers, their higher complexity allow better controllability over the dispersion process by controlling the rotation speed. Wangler et al. [9] outlined the mechanical design of dynamic mixer in terms of the reactor sizing, impeller geometry, motor power and inlet location, which can be used to ensure an appropriate resident time window of chemical admixtures. The most commonly seen dynamic mixer is the screw mixer, which has been implemented across various industrial and academic research bodies [29,38,52,81]. Notice that, although the screw mixer is usually implemented coinciding with the screw extrusion mechanism, there are circumstances where two screws are used separately for extrusion and secondary mixing such as in [29].

The secondary mixing coupled with secondary dosage prevails over the all-in-one mixing (i.e., additives mixed with other concrete constituents simultaneously before pumping) because the latter may associate with additional pumping pressure. In addition, the all-in-one mixing is not favorable for the buildability of FCP, as reported by [129], the reshearing of FCP with pre-mixed additive will significantly reduce its yield stress. The secondary mixing/dosage strategy also allows the possibility of multi-material extrusion [130]. To realize this approach, one typically has to modify the extruder to accommodate additional pipe and pump for feeding secondary materials. Additionally, one also needs to select appropriate types and dosages of chemical additives based on the desired rheological requirements and the pumping distance. Muthukrishnan et al. [124] and Marchon et al. [131] provided a comprehensive summary of the additives that could be added and efficiently dispersed at the extruder.

The secondary mixing coupled with secondary dosage is definitely associated with higher material (additives), capital, and maintenance (inspection and cleaning) costs. The calibration between mechanical parameters, operational parameters, concrete material property, and chemical admixture type and dosage increases the technical complexity. An even higher complexity of dynamic mixer-secondary dosage systems may be achieved in future works that enable extrusion of different materials at different positions and layers, which is dependent on the printing path.

### 4.2. Setting/Fluid on Demand Sub-Processes and Systems Based on External Solicitations

An alternative effective solution to chemical activation (Section 4.1), resolving the conflicts between the pumpability and extrudability with the buildability, is the implementation of the external actions inducing setting-/fluid-on-demand at the extruder.

In the case of setting-on-demand, one can modify the rheology of FCP at a specific time to meet the rheological requirements by employing thermal heating and electro/permanent magnet. The principle of thermal heating rests on the interdependency between the FCP reaction kinetics and temperature and strength development. There are several approaches to heating the extruder chamber: pads and coils, microwave, direct electric curing, ultrasonic pulse, and external heating (e.g., heat gun and lamp). Installing pads or coils within the extruder system to heat concrete is cost-effective. However, from the technical perspective, such an approach is criticized for its [132,133] (1) low heat conduction, which necessitates a large extruder to accommodate a large number of pads or coils and (2) the temperature gradient, which may exacerbate the anisotropic mechanical properties. An alternative approach is microwave-assisted heating, which utilizes a high-energy electromagnetic field to resonate the molecules within FCP to activate molecular vibration, thereby achieving uniform heating [134]. Muthukrishnan et al. [37] found that microwave-assisted heating can accelerate the geopolymer setting process and can improve the stiffness and interlayer bonding strength at appropriate heating duration [135]. The ultrasonic pulse [136] and direct electric curing [137,138,139] have similar working principles, and their enhancements of strength development have been experimentally verified. Lastly, external heating can be provided to FCP extrudate shortly after extrusion. Kazemian et al. [140] found that attaching a heat gun to the extruder system to thermally activate concrete shortly after extrusion can effectively reduce vertical deformation. Bos et al. [141] revealed that the deployment of heat lamps can improve the mechanical properties of concrete after a long exposure time. One point worth noting, emphasized by [124], is that if thermal heating is adopted to fulfill setting-on-demand, it is necessary to account for the thermal effects in both analytical and numerical models for the extrusion process.

The electro/permanent magnet technique was first introduced by [142] for concrete applications. The technique hinges on the principle of magnetorheological fluid (MF) which is produced by incorporating magnetic particles (usually iron or iron-based) of micro-order into a carrier fluid (in this case, FCP). Upon imposition of an external magnetic field by electro/permanent magnets, the magnetic particles within the cementitious MF interact with the field through magnetic dipole alignment in a responsive manner. The interest of this technique in the field of E3DCP is its capability to enhance the yield strength by passing the FCP extrudate through a magnetic field perpendicular to the extrusion direction. The interactions induce the formation of columnar structures that could hinder the particle motion, thereby increasing the yield strength and attaining setting-on-demand [143]. Additionally, a suitable choice of magnetic particles could also enhance the mechanical properties of concrete [143].

An alternative to the setting-on-demand system is the fluid-on-demand system, which reduces the yield stress through vibration. The vibration is a well-known compaction process in the concrete industry to expel entrained air and render concrete with excellent filling and passing ability surrounding the reinforcements [144]. Depending on the frequency, amplitude, vibration time, vibration distance between FCP and vibrator (which depends on the installation location of the vibrator) and vibration volume, the vibration effects can vary [144]. A consensus is that the vibration could reduce the yield stress of FCP and consequently the extrusion pressure, which is particularly of interest for applications in the infinite brick extrusion [24,145]. Sanjayan et al. [146] investigated the hardened concrete properties after vibration. Increases in compressive strengths in all directions and flexural strength were observed for all concrete specimens, which may be due to the pore reduction and refinement from the vibration.

According to [145,146,147,148], some rules of thumb for vibration can be outlined: (1) the vibration frequency should be high enough to reduce the yield stress and prevent the phase separation but not be too high to disturb the shape retention ability of concrete and aspect ratio of FCP extrudate. As highlighted by [146], one can take the critical peak velocity as the reference parameter for determining a suitable frequency; (2) the vibration source is recommended to be installed near the outlet to reduce the vibration distance as the region nearby the outlet is highly prone to blockage and phase separation; (3) an appropriate vibration volume of concrete should be selected, as a large vibration volume of FCP increases the vibration distance and potentially lead to non-uniform vibration; and (4) since the vibration can reduce the FCP yield stress, it is recommended to apply vibration as a set-on-demand method for highly thixotropic material that requires aid during extrusion and can recover strength rapidly after extrusion.

### 4.3. In-Process Reinforcement Sub-Process and System

Considering the well-recognized relatively low tensile strength of concrete, the reinforcement becomes particularly necessary to improve the tensile strength and inhibit the formation of cracks. However, the reinforcement strategies for E3DCP are one of the most critical challenges that hinder its scalability. An effective reinforcement strategy outlined by [61] includes automation potential, fast implementation and wide applicability in different E3DCP scenarios. E3DCP reinforcement can be classified according to the moment of application of reinforcement as pre-process (i.e., reinforcement occurs during the mixing sub-process), single-step process (i.e., in-process reinforcement which occurs during the principal shaping process), and two-step process (i.e., an additional principal reinforcement process takes place either prior to (pre-installed reinforcement) or after (post-installed reinforcement) the principal shaping process) [61]. In Figure 8, a special focus is given to in-process reinforcement methods that have an impact on the extruder system design.

In our case, the advanced fittings of E3DCP system are only limited to the single-step process because (1) the pre-process is not associate with an additional reinforcement sub-process because the reinforcement material (i.e., fiber) is added during the mixing sub-process and (2) the scope of the advanced fittings is only limited to those fittings that bring an additional advanced sub-process into the E3DCP process chain, so pre-installed or post-installed reinforcements—which occur as an additional principal process—are out of scope of the E3DCP advanced fittings. This section will present the advanced fittings of existing or potential reinforcement strategies that can be integrated into the E3DCP system to allow in-process reinforcement based on the E3DCP reinforcement classification framework proposed by [61].

The in-process reinforcement can take one of the following forms [61]: (1) bars; (2) grids, meshes and cages; (3) pre-stressing strands; (4) cables and yarns; (5) textiles; (6) short fibers; and (7) pin and screw. They can be integrated into concrete by entrainment, placing between layers, cross-layer encasement and cross-layer penetration, as shown in Figure 9.

The entrainment of cable/yarn, mesh/textile and short fiber within concrete filament has proven to be an effective reinforcement strategy, see Figure 9a,b. For instance, Bos et al. [149,150] have customized an extruder with a hybrid down/back-flow nozzle that could deposit metal cable-embedded FCP extrudate. This is similar to the continuous fiber-filament for FDM 3D printers [151]. Although the pull-out test and four-point bending test have revealed the inferior mechanical properties of metal-cable-reinforced concrete specimens compared to the conventional bar-reinforced concrete specimens, the concept is proven to be valid as an automated reinforcement strategy. Lim et al. [92] designed an extruder system to embed metal cables within a fiber-reinforced geopolymer composite, enabling a hybrid-reinforcement approach. The result reveals that the hybrid-reinforcement enhances the flexural strength of printed geopolymer specimens by 290% in comparison to printed plain geopolymer specimens. Similar extruder systems have been developed by Neef et al. [152] to embed multiple mineral-impregnated carbon-fiber (MCF) yarns into concrete filament and Li et al. [153] to embed micro-cables in order to create an anisotropic material that can be very efficient in tension and under bending [154,155].

Placing (pre-bent) steel bar [156,157], cable/yarn [158], mesh/textile [29,159] and short fiber between consecutive concrete layers, either manually or automatically, is a common practice in reinforcing E3DCP concrete structure, see Figure 9c,d. Most literature studies or industrial works adopt manual placement in consideration of the cost-effectiveness and ease of operation, which is inevitably associated with printing interruption and placement uncertainty. While some studies [61] advocated the use of augmented reality to improve the accuracy of manual placement, automatic placement could fundamentally circumvent both issues. For instance, Hack et al. [160] and Mechtcherine et al. [161] realized the automatic fabrication of steel bar/mesh/cage, from which one can foresee the prospect of integrating in-situ steel bar/mesh/cage fabrication and placement with the E3DCP process. Automatic placement of cable/yarn was actualized in the work of [158]. Essentially, an MCF yarn is laid down by an auxiliary feeder before the deposition of concrete by the extruder. Mechtcherine et al. [61] pointed out that this method eliminates the interdependency between the reinforcement strategy and design geometry.

The cross-layer encasement entails the partial encasement of the pre-placed bar, mesh/textile, and cage by the concrete filament, see Figure 9e. Mechtcherine et al. [162] proposed to simultaneously deposit concrete layer and steel bar reinforcement using a 3D concrete printer and a wire arc additive manufacturing 3D printer, respectively. Although the authors did not fabricate the actual prototype that hybrids both functionalities, the concept has been validated experimentally with positive feedback. Classen et al. [163] theorized an extruder system that combines the extruder concept from [162] with a fork-shaped nozzle. The extruder consists of a fork-shaped nozzle and a welding unit that enable simultaneous deposition of FCP and the welding of steel meshes in both vertical and horizontal directions. In both cases of [162,163], it is important to coordinate the relatively slow metal welding process with the relatively fast FCP deposition to prevent excessively long layer cycle time. Marchment and Sanjayan [164] proposed an extruder system design that consists of a mesh roller and a fork-shaped nozzle. During printing, vertical steel mesh can be placed at the center of the preceding concrete filament by the roller, which is then partially encased by the concrete released from two directions. This two-part concrete flow favors the formation of strong bonding between the mesh and concrete [164].

As the name implies, cross-layer penetration entails the insert of bar/screw/nail into multiple concrete layers to provide cross-layer reinforcement, see Figure 9f [61]. Some proof-of-concept studies have been carried out. Marchment and Sajanyan [165] used a guide tube to simulate the automated bar-driving process. By testing the produced reinforced specimens, it was shown that the penetration depth is a main factor of the bond strength. The study by [166] suggested that different driving mechanisms affect the bonding strength between the steel bar and concrete. More specifically, screwing the bar into the printed concrete could create stronger bonding than the direct insertion. Wang et al. [167] proposed an in-process U-nail reinforcement system that utilizes magnetic force to insert U-nails vertically to connect just-deposited concrete layer with multiple previously-deposited layers. The penetration of U-nails was shown to improve the interlayer bond strength by 37.8–61.8%, depending on the horizontal spacing between U-nails, vertical penetration depth and U-nail geometries. This type of reinforcement method is close to the nail insertion proposed by Perrot et al. [168], screw insertion developed by Freund et al. [166] or steel rebars penetration as proposed by Hass and Bos [169]. Mechtcherine et al. [61] suggested that the penetration depth should be carefully determined to minimize the disturbance of penetration to the printed layers.

Credit should also be given to fiber-reinforcement, which is the most commonly used reinforcement strategy for E3DCP due to its ease of implementation and cost-effectiveness. It does not require a separate reinforcement sub-process. Various studies have proven its effectiveness in improving the mechanical properties of printed concrete structures, and one crucial determinant is the fiber alignment with the concrete filament after deposition [56,70].

Although fiber-reinforcement is actually a pre-process reinforcement, some advanced fittings can enhance the fiber alignment. For example, Mu et al. [170] and Abavisani et al. [171] presented the imposition of a magnetic electric field to assist the alignment of steel fibers within the concrete, which can be potentially incorporated into the E3DCP extruder system.

### 4.4. Interlayer Bonding Enhancement Sub-Process and System

The presence of interfaces with specific bonding strength is an intrinsic characteristic of E3DCP that can greatly impact the overall bulk strength and durability of the E3DCP structures that have been validated by various studies [96,172]. The latter motivates the seeking of interlayer enhancement strategies that can guarantee minimal bonding strength and mechanical properties. This is particularly necessary in the case of on-site printing where the material undergoes outside air velocity and drying that can be the origin of cold joint and material anisotropy [172].

Kruger and Zijl [173] epitomized some possible sources of interlayer bond weakness including moisture loss, air entrapment, thixotropy and surface roughness. To minimize the adverse effects of these sources, apart from manipulating the material design and operational design (e.g., layer cycle time), a novel strategy is to deposit bonding agents between consecutive concrete layers to create an interface bonding layer. Studies have investigated different bonding agents, and they are typically cement paste/mortar consisting of chemical additives [174], sulfur-black carbon [175], epoxy resin-based and chloroprene latex-based polymer [176], all of which show different extents of improvements in interlayer bond strength. Most of the studies have not realized the automation of bonding agent deposition, but the associated technical barrier can be readily overcome. Weng et al. [98] have devised an automated bonding agent deposition system alongside the extruder system. During the deposition process, the bonding agent spray moves ahead of the printing nozzle and covers the preceding concrete layer with the bonding agent, which is overlayed by a new concrete layer shortly. Marchment et al. [174] and Verian et al. [52] adopted a similar deposition method to minimize the interlayer porosity and enhance interlayer bond strength.

Physical means are also effective in enhancing the interlayer bonding. Keita et al. [172] highlighted the significant role of moisture loss in reducing the interlayer bonding strength. The excessive moisture loss can induce a dry front with a thickness around several hundreds of micrometers at top of a concrete layer. It was suggested the remixing of this dry front with the other portions of the concrete layer can achieve rehomogenization and cancel the adverse consequences of moisture loss, thereby inhibiting the formation of weak interlayers [172]. Hence, the incorporation of mechanical remixing fittings within the extruders can be promising. Several studies investigated a topological interlocking strategy, which relies on creating grooves on the surface of the printed layers that are expected to interlock layers and improve their adhesion and the mechanical behavior of the printed structures [177,178,179]. It was shown that the incorporation of topological interlocking could improve the interlayer bonding strength by 26% [178]. Similar positive results have been reported by [179]. Nevertheless, the technical barrier still exists in the incorporation of relevant hardware for automation, as the studies mentioned before fabricated the interlocking configurations using mold casting. Lastly, the aforementioned interlayer reinforcement by penetration [167] can also enhance the interlayer bonding strength.

### 4.5. Finishing Sub-Process and System

Due to the nature of the extrusion associated with the E3DCP process, the printed structures are generally characterized by layer-by-layer patterns that can adversely affect aesthetic aspects. A finishing system can be implemented during the principal process to improve the pattern. In the contour crafting pioneered by [180,181], a trowel is added to constrain the flow of concrete filament in the y-direction to improve the surface finish. Souza et al. [182] advocated the combination of a side trowel for a circular nozzle to smooth the rounded extrudate. The studies in this respect are essential as the structural aspects of printed structures become more understood.

### 4.6. Support Placement Sub-Process and System

The support placement is required when an overhang exists in the printed structure such as the holes for windows and doors. Hoffmann et al. [91] developed a printing system that incorporates the grasping functionality, enabling automatic placement of support structures during the 3DCP process. When the extruder moves to the position where support is needed, the concrete extrusion pauses and the grasping device is activated to place a lintel at the designated position, after which the concrete extrusion and deposition resume. The merit of such an approach is still in question, as the coupling of concrete extrusion and grasping necessitates a temporary pause of one task to allow the performance of the other task, which is not favorable for creating a seamless 3DCP process. Additionally, in large-scale projects where the placement of supports could take a long time, the interlayer bonding between concrete layers can be adversely affected. One can resolve such concerns using a team of robots [119].

### 4.7. Monitoring and Feedback Sub-Process and System

The inclusion of a monitoring and feedback system in the printing system can improve the robustness of E3DCP. For instance, it is well-known that pumping can lead to process-induced variations of concrete rheology. To address the issue, Ji et al. [29] customized an extruder equipped with a servo motor that can measure the torque resistance of FCP during the extrusion process, thereby allowing inline real-time rheology characterization. The study suggested that a suitable printable concrete mix design has a slump of 120–130 mm, which corresponds to the torque resistance values of 1.25–1.75 Nm of the servo motor. If the FCP within the extruder falls out of this range, additional concrete constituents are added through an additional feeding system within the chamber to adjust the rheology. Additionally, apart from the motor loading, motor temperature and printing speed are also monitored, and in the case of overloading, an emergency stop can be executed to pause the printing process [29].

The study of [25] highlighted the adverse effect of material heterogeneity on the extrusion rate consistency. It was suggested to install an extrusion rate sensor at the extruder to provide real-time feedback and respective adjustment. The research team from Eindhoven University of Technology [183] emphasized that, during printing, the actual standoff distance (the distance between the bottom of the extruder and the printing surface) can deviate from the prescribed values due to the positioning inaccuracy, printing surface unevenness as well as progressively increasing vertical deformation during the E3DCP process, which potentially leads to impaired mechanical properties [82]. Accordingly, the team developed a device attached to the extruder that could enable self-leveling of the extruder. The actual standoff distance measured by the device is used to calculate the height deviation in real-time, according to which the extruder position can be adjusted.

Kazemian et al. [184] developed a monitor that allows the detection of flow deposition rate. Essentially, the monitor consists of a high-resolution camera and a Raspberry Pi 3 model B microprocessor and, during the deposition, it could keep track of the layer width in real-time and, in the case of over-extrusion or under-extrusions, the printing operational parameters are adjusted automatically.

## 5. Discussion

### 5.1. Process Chain of E3DCP System

The choices of printing system, basic fittings and advanced fittings determine the relations between the principal process and sub-processes (i.e., in series, simultaneous and contiguous) [1] as well as the nature of the principal process and subprocess (i.e., continuous or cyclic), as shown in Table 9. Buswell et al. [1] presented a method of pictorial representation of the process chain of any DFC technology. Accordingly, one can produce the process chain for any E3DCP system. Figure 10 illustrates the process chains of three different E3DCP systems.

The basic fittings are relatively well-understood through decades of practice in concrete construction and, with the rise of popularity of E3DCP in the recent decade, considerable research attention has been paid to the printing system to understand the underlying mechanisms and outline the design protocol. However, the understanding of each advanced fitting is relatively poor, and most E3DCP systems so far can only integrate one or two of the advanced fittings. The integration of advanced fittings into the E3DCP system still face several bottlenecks: (1) the design standard of each advanced fittings needs to take account of its compatibility with the principal process (e.g., support placement and deposition) so that no or minimal adverse effects are induced; (2) the compatibility between different advanced fittings (e.g., finishing system and support placement, secondary mixing and in-process reinforcement) needs to be resolved; and (3) with more advanced fittings being integrated into the E3DCP system, it is difficult to find the optimal sequence of advanced sub-processes due to increasing complexity of the process chain.

### 5.2. E3DCP Classification Framework

Based on Section 2 and Section 3 and the notation classification framework of [74], this section intends to propose a new notation classification framework specifically for E3DCP as an extension of the DFC classification framework by [1] (i.e., at the level of material extrusion). The classification framework consists of 12 parameters relating to the environment, application use and principal process mechanical system as shown in Table 10. Table 11 summarizes some examples of E3DCP technologies based on this framework.

Several remarks need to be made: (1) this framework is established based on the mechanical systems of E3DCP principal processes (i.e., the printing system of the extrusion and deposition processes), therefore the sub-processes do not affect the classification, which complies with the philosophy of [1]; (2) this framework excludes the smart dynamic casting [7] that is classified as formative instead of additive by [1], even though it is included in the classification framework of [74]; (3) the “environment” and “application use” in Figure 1 are now part of the notation representation, see Table 10; (4) some parameter aforementioned in Section 2 (i.e., extruder wall roughness and chamber diameter and length) are not included in this framework due to lack of specifications in the literature, but it should not be neglected in the future work; (5) some parameters from [74] are still adopted in this framework, but with a different notation: “object scale” x_o_ is replaced by “outlet exit size” O_d_, “extrusion scale” x_e_ is replaced by “outlet exit size” P_b_, “environment” e is replaced by “environment” E, “robotic complexity” x_o_ is replaced by “degree of freedom” P_d_; (6) the “assembly parameter” from [74] is not considered because it involves a different principal assembly process other than the principal shaping process; and (7) the “support parameter” from [74] is not considered because support placement is an advanced sub-process which should not affect the classification [1].

## 6. Conclusions

The paper provided a comprehensive review of the mechanical design of the E3DCP principal shaping process (i.e., the extruder system which includes the general extruder design, chamber design, outlet design, and the positioning system) and advanced sub-process (i.e., advanced fittings). Accordingly, the following conclusions can be reached:The concrete extrusion process originates from the competition between the extrusion drive force, Fe and extrusion resistive forces, which may include chamber wall shear force F_cf_, shaping force F_pl_, nozzle wall shear force F_nf_, dead zone shear force F_df_ and layer pressing force, F_lp_;The three possible extrusion mechanisms—primary motivation, ram extrusion and screw extrusion—provide pumping and gravity force F_pg_, ram extrusion force F_ram_ and screw extrusion force F_screw_, respectively;A low extruder wall roughness can reduce F_cf_ and F_nf_, thereby reducing the extrusion pressure;The chamber design needs to consider chamber number, chamber length and diameter, and chamber tapering. A smaller chamber diameter increases F_pl_; and the chamber tapering can generally reduce F_pl_ and extrusion pressure;The outlet design needs to consider outlet orientation, outlet form, outlet tapering, outlet cross-section shape, and outlet exit size. The outlet form of the orifice is associated with lower extrusion pressure whereas the outlet form of the nozzle has higher extrusion pressure due to the presence of F_nf_; the presence of outlet tapering increases F_nf_; the circular cross-section more likely induces a free flow extrusion mode, whereas the rectilinear cross-section more likely induces an infinite brick extrusion mode; a smaller outlet exit size corresponds to enhanced F_pl_ and F_nf_;The advanced fittings include the secondary mixing, setting-/fluid-on-demand, in-process reinforcement, interlayer bonding enhancement, finishing, support placement, and monitoring and feedback processes. They are still at a nascent stage of application in E3DCP systems, and the incorporation of advanced fittings could increase the complexity of the E3DCP process chain, requiring more investigations in the respects of: (a) the compatibility between each advanced fitting and the printing system; (b) the compatibility between different advanced fittings; and (c) the optimal sequence of advanced sub-processes;The most crucial aspect of the E3DCP extruder system is the understanding of how the coupling between the mechanical designs, different concrete materials (e.g., low- and high-viscosity, low- and high-solid concentration), and operational design (e.g., pumping pressure) can influence the extrusion forces and phenomena, from which one can draw guidelines for the corresponding mechanical system and material combinations that can optimize the extrudability. As one can tell, considerable research efforts are required to fully understand this chain effect: the coupling of mechanical and material designs, the competition of extrusion forces, the occurrence of various extrusion phenomena, and the extrudability of the overall setting.

In addition, this paper identifies the principal process, basic sub-processes, and advanced sub-processes of the E3DCP process chain, based upon which a notation classification framework of the E3DCP system was proposed as an extension to the DFC classification framework by [1]. The authors reckoned that such a classification framework could assist a more systematic E3DCP printing system design. Considering the nascency of E3DCP, some of the mechanical aspects (e.g., extruder wall roughness and chamber diameter) have not yet been taken into account in the classification framework. The prospective E3DCP literature should provide specifications of the mechanical parameters (e.g., D_c_, D_entry_, D_exit_ and L_n_) to establish a database for various forms of research such as machine learning.

## Figures and Tables

**Figure 1 materials-16-02661-f001:**
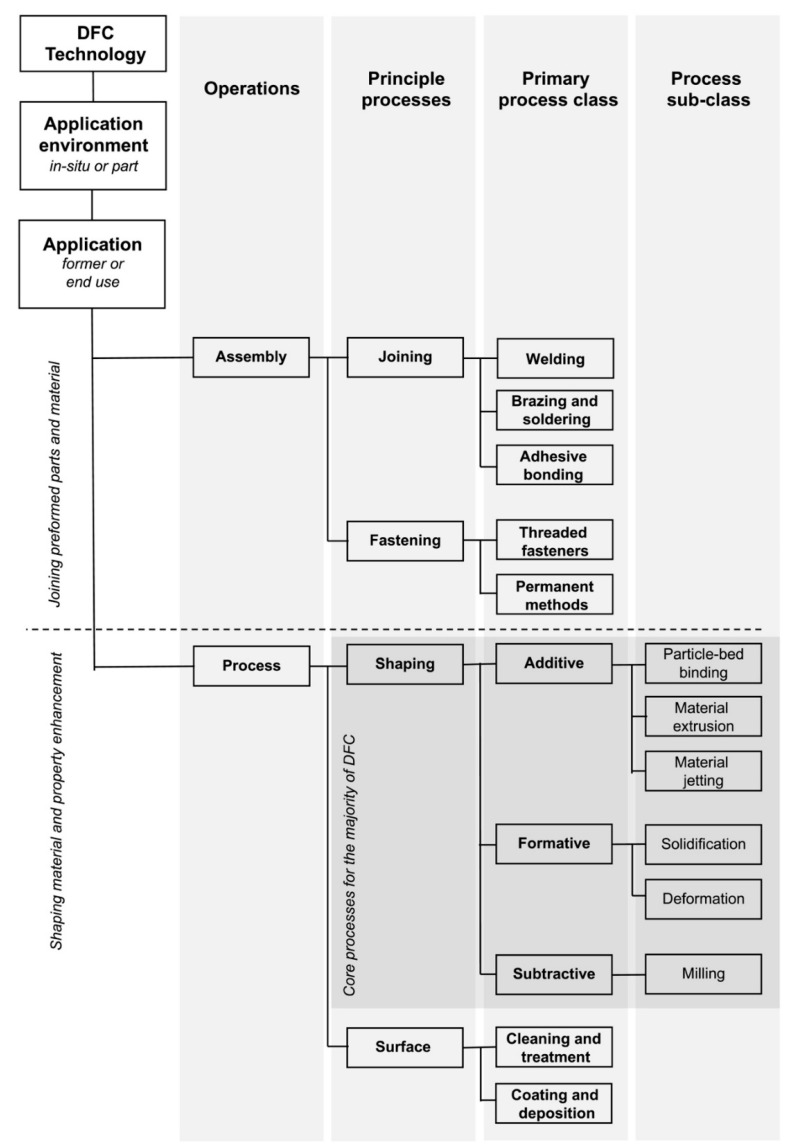
The process classification framework of DFC technologies proposed by [1].

**Figure 2 materials-16-02661-f002:**
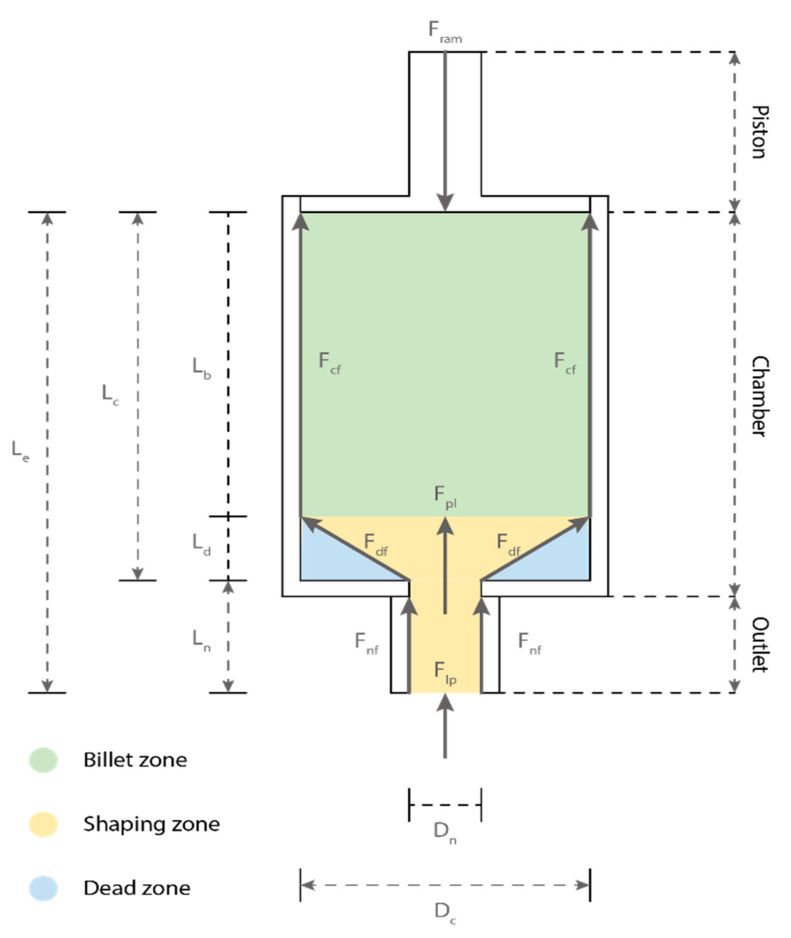
The printhead of a typical ram extrusion for E3DCP.

**Figure 3 materials-16-02661-f003:**
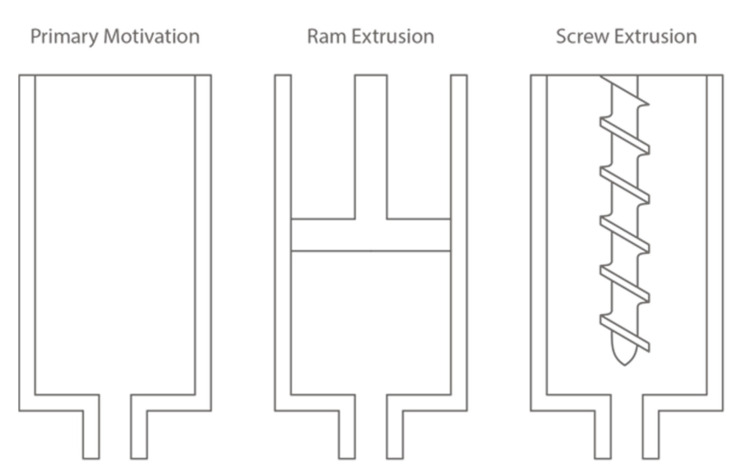
The schematic sketches of extruder mechanisms: primary motivation, ram extrusion and screw extrusion.

**Figure 4 materials-16-02661-f004:**
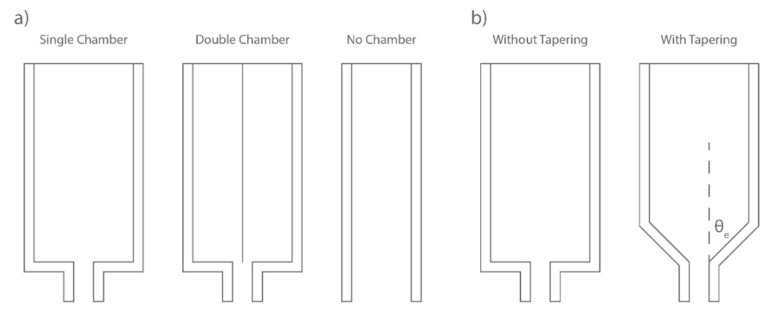
The schematic sketches of (**a**) chamber number: single chamber, double chamber and no chamber; and (**b**) chamber tapering: without tapering and with tapering.

**Figure 5 materials-16-02661-f005:**
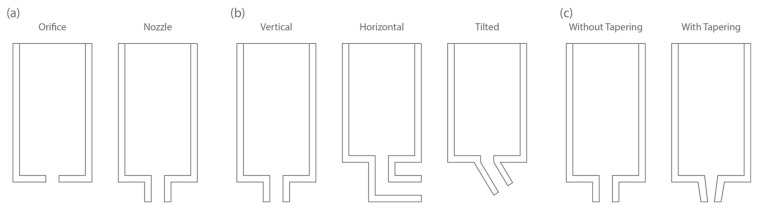
The schematic sketch of (**a**) outlet form: orifice and nozzle; (**b**) outlet orientation: vertical, horizontal and tilted; and (**c**) outlet tapering; without tapering and with tapering.

**Figure 6 materials-16-02661-f006:**
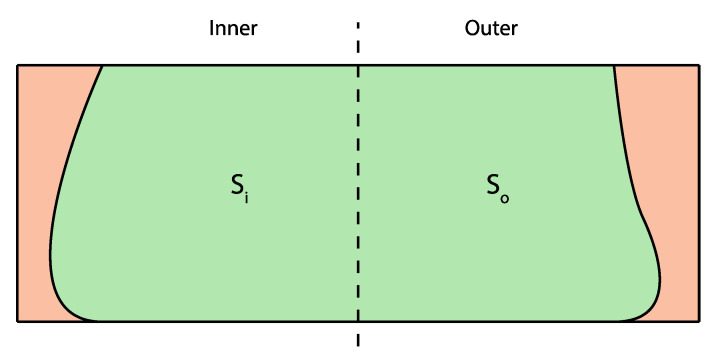
The illustration of the mass distribution ratio [71].

**Figure 7 materials-16-02661-f007:**
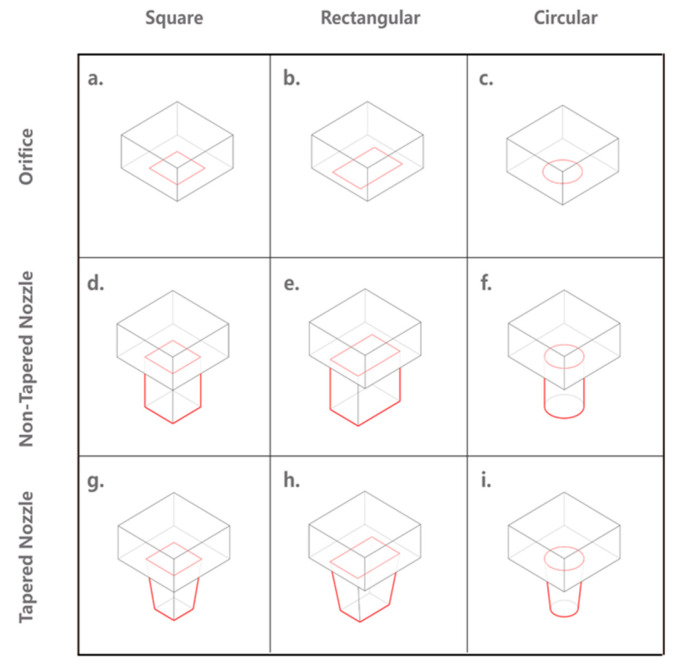
Outlets of different forms (orifice and nozzle), outlet tapering and cross-section shapes: (**a**–**c**) square, rectangular and circular orifices; (**d**–**f**) square, rectangular and circular non-tapered nozzles; and (**g**–**i**) square, rectangular and circular tapered nozzles.

**Figure 8 materials-16-02661-f008:**
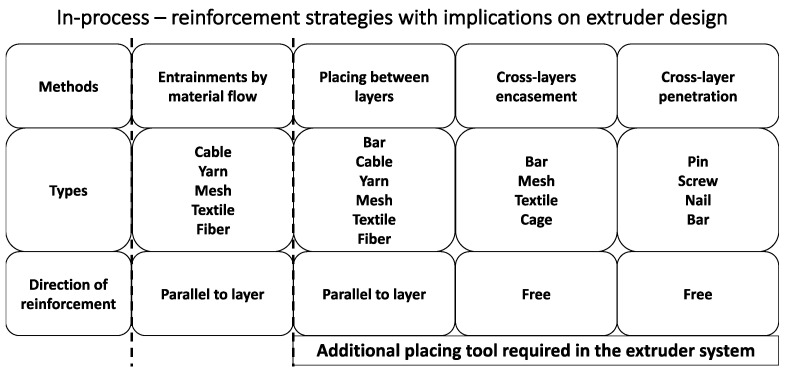
In-process reinforcement methods with implications on the extruder systems according to the process classification framework for integration of reinforcement into DFC technologies (PC4IR-DRC) [61].

**Figure 9 materials-16-02661-f009:**
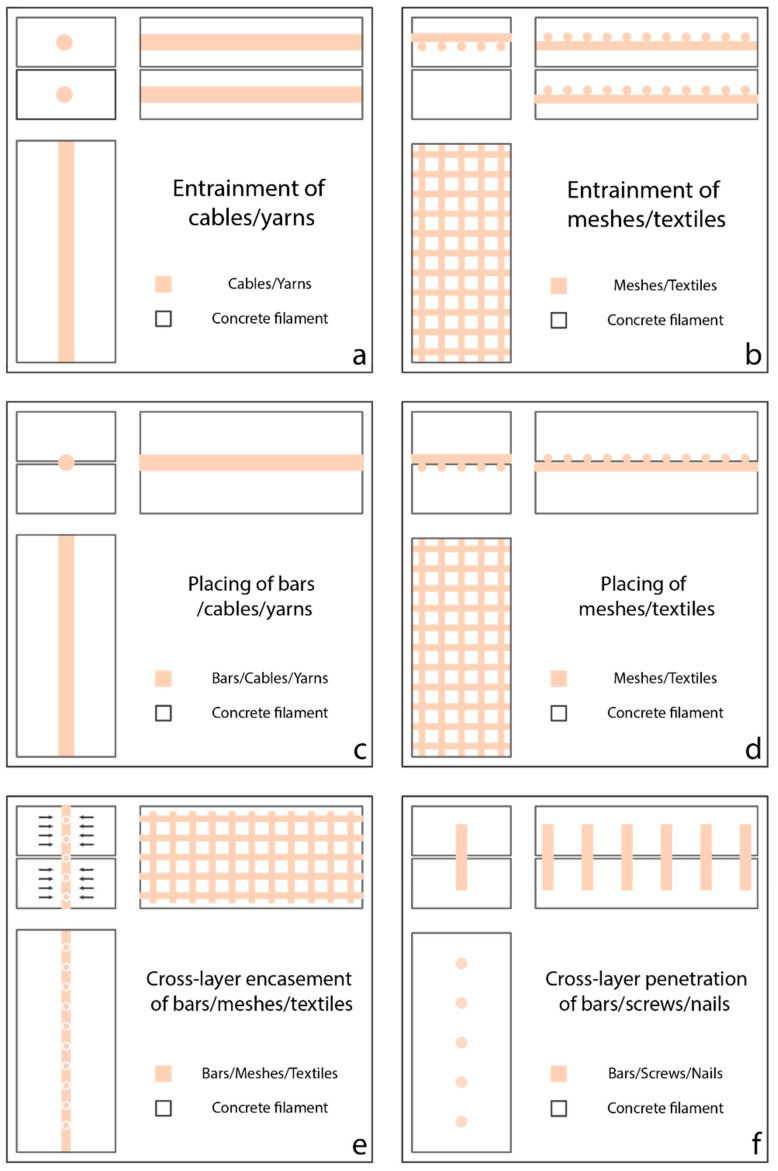
Cross-section view (left-top), side view (right-top) and plan view (left-bottom) of: (**a**) entrainment of cables/yarns; (**b**) entrainment of meshes/textiles; (**c**) placing of bars/cables/yarns; (**d**) placing of meshes/textiles; (**e**) cross-layer encasement of bars/meshes/textiles; and (**f**) cross-layer penetration of bars/screws/nails.

**Figure 10 materials-16-02661-f010:**
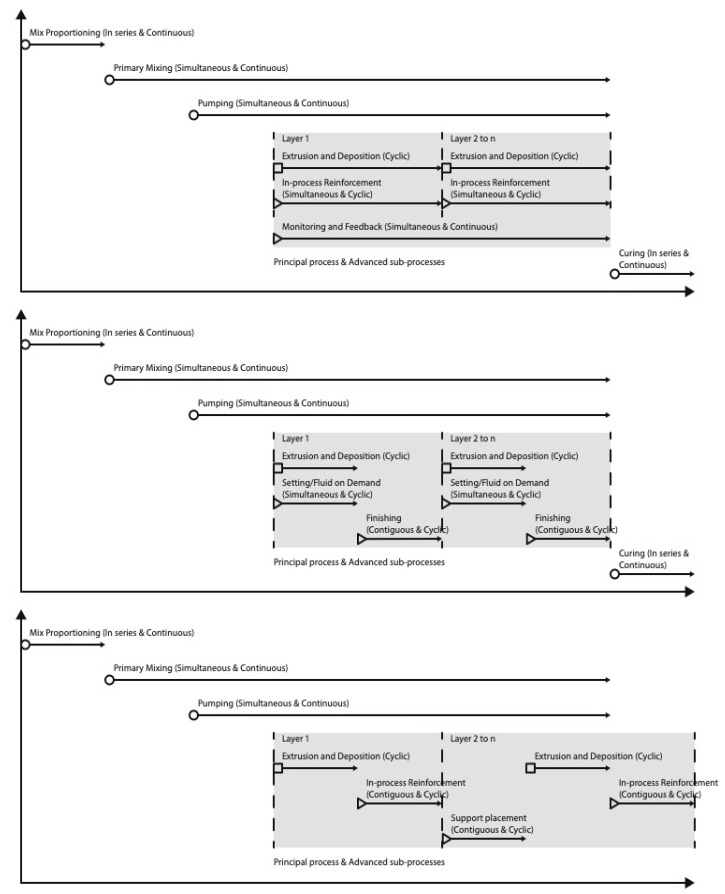
The process chain of three different E3DCP systems of various process combinations according to the pictorial representation method proposed by [1].

**Table 1 materials-16-02661-t001:** E3DCP mechanical system.

	Mechanical System	
Principal shaping process	Printing system	Extruder system
		Positioning system
Basic sub-process	Basic fittings	Mix proportioning system
		Primary mixing system
		Pumping system
		Curing system
Advanced sub-process	Advanced fittings	Secondary mixing system
		Setting-/Fluid-on-demand system
		In-process reinforcement system
		Interlayer bonding enhancement system
		Finishing system
		Support placement system
		Monitoring and feedback system

**Table 2 materials-16-02661-t002:** The effects of extruder mechanisms on the extrusion forces, extrusion behaviors, extrudability and economic aspects and technical complexity based on the same material design and operational design.

		Primary Motivation	Ram Extrusion	Screw Extrusion
Extrusion forces	Primary extrusion driving forces	• F_pg_	• F_ram_	• F_screw_
	Extrusion resistive forces	• F_pl_ • F_cf_ • F_df_ • F_nf_ • F_lp_		
Extrusion behaviors		• Low shearing • Susceptible to over-extrusion due to inertia	• Higher shearing • Increase the consolidation and phase separation and the dead zone formation • Increase the risk of blockage • Higher extrusion pressure required	• Higher shearing • Reduce the consolidation and phase separation And the dead zone formation • Reduce the risk of blockage
Extrudability		• Largely dependent on the materials	• Lower extrudate homogeneity	Smooth extrusion • Higher extrudate homogeneity • Lower shape retention unless combined with secondary dosage
Economic aspects and technical complexity		• Lower energy consumption • Lower capital and maintenance costs • No additional technical complexity	• Higher energy consumption • Higher capital and maintenance costs (a higher risk of blockage) • Additional technical complexity due to the calibration of the ram extruder mechanical design and operational parameters with respect to concrete material properties to prevent phase separation and minimize property inconsistency in multiple extrudates	• Higher energy consumption • Higher capital and maintenance costs • Additional technical complexity due to more mechanical design parameters (e.g., screw dimensions) are involved, which require calibration with respect to the concrete material properties

**Table 3 materials-16-02661-t003:** The effects of extruder wall roughness on the extrusion forces, extrusion behaviors, extrudability and economic aspects and technical complexity based on the same material design and operational design.

	High Surface Roughness (Ra)	Low Surface Roughness (Ra)
Extrusion resistive forces	• Higher F_cf_ and F_nf_	• Lower F_cf_ and F_nf_
Extrusion behaviors	• Higher shearing • Increase the consolidation and phase separation and dead zone formation • Higher risk of blockage • Higher extrusion pressure required	• Lower shearing • Reduce the consolidation and phase separation and dead zone formation • Lower risk of blockage • Lower extrusion pressure required
Extrudability	• Less smooth extrusion • Lower extrudate homogeneity	• More smooth extrusion • Higher extrudate homogeneity
Economic aspects and technical complexity	• Higher energy consumption to overcome F_cf_	• Lower energy consumption • Higher capital costs if polishing and lubrication are applied. The lubrication may be associated with higher maintenance cost.

**Table 4 materials-16-02661-t004:** The effects of chamber tapering on the extrusion forces, extrusion behaviors, extrudability and economic aspects and technical complexity based on the same material design and operational design.

	High Outlet Entry Angle	Moderate Outlet Entry Angle
Extrusion resistive forces	• Higher F_pl_	• Lower F_pl_
Extrusion behaviors	• Higher shearing • Increase the consolidation and phase separation and dead zone formation • Higher risk of blockage • Higher extrusion pressure required	• Lower shearing • Reduce the consolidation and phase separation and dead zone formation • Lower risk of blockage • Lower extrusion pressure required
Extrudability	• Less smooth extrusion • Lower extrudate homogeneity	• More smooth extrusion • Higher extrudate homogeneity
Economic aspects and technical complexity	• Higher energy consumption compared to moderate tapering • Higher capital costs compared to no tapering • Additional technical complexity	• Lower energy consumption compared to high tapering • Higher capital costs compared to no tapering • Additional technical complexity

**Table 8 materials-16-02661-t008:** The material costs and technical complexity of the advanced fittings.

Advanced Fittings		Material Cost	Technical Complexity *	
Secondary mixing system (with secondary dosage)	Static mixer	• Higher (additives)	• Low	• The compatibility of different static mixers with different concrete materials.
	Dynamic mixer	• Higher (additives)	• Medium/High	• The optimization of mechanical parameters, operational parameters, concrete material property, chemical admixture type and dosage and printing path.
Setting/Fluid on demand system	Thermal heating	• Non	• Low/Medium/High *	• Thermal gradients that can lead to non-uniform modifications of concrete properties. • Numerical modelling of the thermal effects during concrete extrusion.
	Electro/permanent magnet	• Higher material (magnetic particles)	• Medium/High *	• Compatibility of magnetic particles with concrete materials. • The guidelines for operational parameter control.
	Vibration	• Non	• Medium/High *	• Impacts of vibration on the material extrudability.
In-process reinforcement system	Entrainment	• Higher (reinforcements)	• Medium/High *	• The control of the feed-in speed of the reinforcement materials. • The correct alignment of the reinforcement with respect to the concrete layer cross-sectional centroid to prevent anisotropic properties and ensure uniform covering
	Placing between layers		• High/High *	• Concrete materials with appropriate rheological properties to seal the horizontal weak interface which would be otherwise susceptible for moisture and chemical invasions. • Precise positionings of the reinforcement
	Cross-layer encasement		• High/High *	• Concrete materials with appropriate rheological properties to seal both the vertical and horizontal weak interfaces • Precise positionings of the reinforcement in terms of the centerline alignments.
	Cross-layer penetration		• High/High *	• Precise positionings of the reinforcement in terms of the spacing and centerline alignments.
Interlayer bonding enhancement system	Bonding agents	• Higher (bonding agents)	• Medium	• Compatibility of the bonding agents with the concrete materials.
	Physical	• Non	• Medium/High	• The implementations of the physical means without affecting the extrusion process.
Finishing system		• Non	• High	• More precise precision according to the printing path
Support placement system		• Higher (supports)	• High	• Precise positions of the supports. • The effects of pause on the printing time and open time of the concrete materials.
Monitoring and feedback system		• Non	• Medium/High	• The monitoring itself is not complex, however, the real-time analysis, feedback and adjustment can significantly increase the complexity

Low, when the system is a passive system; medium, when the system is automated but independent of the printing path and programming; high, when the system needs to be integrated and programed with the printing path definition to perform its intended task; high *, when the system could be coupled with the printing path to achieve functional-graded materials.

**Table 9 materials-16-02661-t009:** The relation between and nature of the E3DCP principal process and sub-processes.

Mechanical System	Process		Relation with the Principal Shaping Process (In Series/Simultaneous/Contiguous)	Continuous/Cyclic
Printing system	Principal shaping process	Extrusion	-	• Cyclic
		Deposition		
Basic fittings	Basic sub-process	Mix proportioning	-	• Cyclic in batch mixing
		Primary mixing	-	• Continuous in continuous mixing • Cyclic in batch mixing
		Pumping	Simultaneous (but occurs earlier) with the principal shaping process	• Continuous
		Curing	In series with the principal shaping process	• Continuous
Advanced fittings	Advanced sub-process	Secondary mixing	Simultaneous with the extrusion process	• Continuous
		Setting/Fluid on demand		
		In-process reinforcement	Simultaneous or contiguous with the principal shaping process	• Cyclic
		Interlayer bonding enhancement		
		Finishing		
		Support placement	Contiguous with the principal shaping process (starting from 2nd layer)	• Cyclic
		Monitoring and feedback	Simultaneous with the principal shaping process	• Continuous

**Table 10 materials-16-02661-t010:** The classification framework of E3DCP based on 12 parameters relating to the environment, application use, principal process mechanical system.

Parameter				Notation	Division
Environment				$E^{0}$	On-site/In-situ (direct printing)
				$E^{1}$	Part in mini-factory/lab
				$E^{2}$	Part in prefabrication factory
Application use				$A^{0}$	End use
				$A^{1}$	Former
Principal process mechanical system	Positioning system		Degree of freedom	$P_{d}^{0}$	One 3-axis robot (gantry)
				$P_{d}^{1}$	One 4-axis robot (gantry with a rotational DOF)
				$P_{d}^{2}$	One 6-axis robot (robotic arm)
				$P_{d}^{3}$	One 6-axis robot on a rail
				$P_{d}^{4}$	One 6-axis robot on a mobile base
				$P_{d}^{5}$	One delta robot
				$P_{d}^{6}$	One swarm robot
				$P_{d}^{7}$	One 6-axis robot on a 3-axis robot
				$P_{d}^{8}$	Multiple 6-axis robots
				$P_{d}^{9}$	Multiple 6-axis robots on rails
				$P_{d}^{10}$	Multiple 6-axis robots on mobile bases
				$P_{d}^{11}$	Multiple swarm robots
			Build volume	$P_{b}^{0}$	Dimension < 1 m
				$P_{b}^{1}$	1 m < Dimension < 4 m
				$P_{b}^{2}$	5 m < Dimension < 10 m
				$P_{b}^{3}$	> 10 m
	Extruder system	General extruder design	Extruder mechanism	$G_{m}^{0}$	Pumping pressure
				$G_{m}^{1}$	Ram extrusion
				$G_{m}^{2}$	Screw extrusion
		Chamber design	Chamber number	$C_{n}^{0}$	One chamber
				$C_{n}^{1}$	Multiple chambers
				$C_{n}^{2}$	No chamber (or uniform with the outlet)
			Chamber tapering	$C_{t}^{0}$	Without tapering
				$C_{t}^{1}$	With tapering
		Outlet design	Outlet form	$O_{f}^{0}$	Orifice
				$O_{f}^{1}$	Single nozzle/die
				$O_{f}^{2}$	Multiple nozzles/dies
			Outlet orientation	$O_{o}^{0}$	Vertical
				$O_{o}^{1}$	Horizontal
				$O_{o}^{2}$	Tilted
			Outlet tapering	$O_{t}^{0}$	Without tapering
				$O_{t}^{1}$	With tapering
			Outlet cross-sectional shape	$O_{s}^{0}$	Square/Rectangular
				$O_{s}^{1}$	Circular
				$O_{s}^{2}$	Elliptical
				$O_{s}^{3}$	Irregular (e.g., hollow-core, U-shape)
				$O_{s}^{4}$	Adjustable
			Outlet exit size	$O_{d}^{0}$	Size < 8 mm
				$O_{d}^{1}$	8 mm < Size < 5 cm
				$O_{d}^{2}$	5 cm < Size < 30 cm
				$O_{d}^{3}$	Size > 30 cm
				$O_{d}^{4}$	Adjustable

**Table 11 materials-16-02661-t011:** Some examples of E3DCP technologies based on this classification framework.

References	Extruder
[89]	$E^{1}A^{0}P_{d}^{2}P_{b}^{0}G_{m}^{0}C_{n}^{0}C_{t}^{0}O_{f}^{1}O_{o}^{0}O_{t}O_{s}^{0}O_{d}^{0}$
[62]	$E^{0}A^{0}P_{d}^{0}P_{b}^{3}G_{m}^{0}C_{n}^{0}C_{t}^{0}O_{f}^{1}O_{o}^{1}O_{t}^{0}O_{s}^{0}O_{d}^{2}$
[38]	$E^{1}A^{0}P_{d}^{0}P_{b}^{0}G_{m}^{2}C_{n}^{0}C_{t}^{1}O_{f}^{1}O_{o}^{0}O_{t}^{0}O_{s}^{0}O_{d}^{1}$
[149,150]	$E^{0}A^{0}P_{d}^{1}P_{b}^{2}G_{m}^{0}C_{n}^{0}C_{t}^{0}O_{f}^{1}O_{o}^{0}O_{t}^{0}O_{s}^{3}O_{d}$
[81]	$E^{1}A^{0}P_{d}^{0}P_{b}^{0}G_{m}^{2}C_{n}^{0}C_{t}^{0}O_{f}^{1}O_{o}^{0}O_{t}^{0}O_{s}^{0}O_{d}^{4}$

## Data Availability

Not applicable.

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
