# Peer review of "A Review of the Extruder System Design for Large-Scale Extrusion-Based 3D Concrete Printing"

_materials, 2023, doi:10.3390/ma16072661_

Round 1
Reviewer 1 Report
A review paper should not only summarise recently published works but also should contain critical and comprehensive discussions. Therefore, check the writing for the whole manuscript. The review should not be presented by listing what others have done.
The title does not correspond to the type of the manuscript (Article - A Critical Review of the ......)
The title does not correspond to the type of the manuscript (Article - A Critical Review of the ......)
Author Response
A review paper should not only summarise recently published works but also should contain critical and comprehensive discussions. Therefore, check the writing for the whole manuscript. The review should not be presented by listing what others have done.
The title does not correspond to the type of the manuscript (Article - A Critical Review of the ......)
We want to thank the reviewer for his comment. We try to add discussions and tables at the end of each section related to technical, economical and environmental impacts of each extruder design. We also removed the word “critical” from the title.

Reviewer 2 Report
The paper presents a review for extrusion based 3D concrete printing. I highly recommend accepting the manuscript for the need of such review paper. The paper is well-written and the figures are clear and descriptive.
I do not have comments on the manuscript and I congratulate the authors for such work.
Author Response
We want to thank the reviewer for his positive feedback on the paper.
Reviewer 3 Report
This manuscript is highlighted on the extruder system design for large scale extrusion-based 3D concrete printing which are followed by the different authors around the world. Although, the authors have summarized the different extruder system well but the scientific value of this manuscript is very low. However, this can be considered as a technical report.
In many sections the authors have just discussed the different parameters used by the researchers and not discussed the impact of those parameters to the 3D printing objects.
It is recommended to summarize the environmental and economic aspects of the different extruder systems in a table.
Author Response
We want to thank the reviewer for his comment. We try to add discussions and tables at the end of each section related to technical, economical and environmental impacts of each extruder design. We also removed the word “critical” from the title. We do hope it now meets sufficient scientific value to this review.

Reviewer 4 Report
Manuscript is a extensive review study based on references of most significant authors in the world.
Currently, significance research efforts are dedicated to the material design to Improve the extrudability of fresh concrete.
There is still a lack of a review paper that highlights the significance of the mechanical design of the E3DCP system.
Concrete material extrusion, referred to as extrusion-based 3D concrete printing (E3DCP) in this paper, has been appreciatled by academia and industry as the most plausible candidate for prospective concrete concretion.
The research relating to the principal process and the advanced sub-processes is scarce due to the fact that are rarely applied to traditional concrete construction projects.
The manuscript pays attention to the most important aspects of the main process and advanced sub-processes for E3DCP applications.
Conclusions do not introduce new, but they are a summary of the entire review article.
I assess the whole Manuscript positively due to:
- extensive review of the current references;
- organized presentation of the most important elements regarding the principal process and advanced sub-processes for E3DCP applications.
Positive comments:
Manuscript does not introduce significant novelties in the subject he describes, but is a very reliable review of the existing state of knowledge.
Critical remarks:
1.
Chapter 2.1.1 and 2.1.2 relate to "Extrusion Mechanism" and have the same subtitle.
I suggest to leave one chapter :
2.1. "General Extruder System Design".
Alternatively, you can do two subsections, but under different subtitles, e.g.
2.1.1. Impact of extruder mechanisms on the extrusion forces
2.1.2. Impact of extruder wall roughness on the extrusion forces
2.
Descriptions on some drawings are not very legible, in particular: Figure 2 and 5.
Author Response
Critical remarks:
1.
Chapter 2.1.1 and 2.1.2 relate to "Extrusion Mechanism" and have the same subtitle.
I suggest to leave one chapter :
2.1. "General Extruder System Design".
Alternatively, you can do two subsections, but under different subtitles, e.g.
2.1.1. Impact of extruder mechanisms on the extrusion forces
2.1.2. Impact of extruder wall roughness on the extrusion forces
We have choose the second option
- Descriptions on some drawings are not very legible, in particular: Figure 2 and 5.
We tried to improve the design of figures 2 and 5.
Reviewer 5 Report
The document is a very comprehensive review paper. This reviewer has only a few minor review requests.
Type of the paper
Since this is a review paper, why was it submitted as an article?
Title
This reviewer did not actually find many points of critical analysis in this paper. Even the conclusions do not seem to be the result of a critical analysis but a collection of conclusions drawn from other papers. If not, Authors should clearly specify what their critical contribution consists of, both in the introduction and in the conclusions. If not, they should change the title from “A Critical Review” to “A Review”.
Figure 6
Please improve the quality of this figure.
Author Response
Title
This reviewer did not actually find many points of critical analysis in this paper. Even the conclusions do not seem to be the result of a critical analysis but a collection of conclusions drawn from other papers. If not, Authors should clearly specify what their critical contribution consists of, both in the introduction and in the conclusions. If not, they should change the title from “A Critical Review” to “A Review”.
We have change the name of the paper as suggested.
Figure 6
Please improve the quality of this figure.
We have changed the name of paper and also redraw figure 6.
Round 2
Reviewer 1 Report
Accept in present form